# Inherently Explainable Reinforcement Learning in Natural Language

**Xiangyu Peng**[†]     **Chen Xing**[†]     **Prafulla Kumar Choubey**[‡] **Chien-Sheng Wu, Caiming Xiong**

[†]Georgia Institute of Technology     [‡]Allen Institute for AI

{xpeng62,riedl}@gatech.edu, raja@allenai.org

## Abstract

We focus on the task of creating a reinforcement learning agent that is inherently explainable—with the ability to produce immediate local explanations by thinking out loud while performing a task and analyzing entire trajectories post-hoc to produce temporally extended explanations. This Hierarchically Explainable Reinforcement Learning agent (HEX-RL),[1] operates in Interactive Fictions, text-based game environments in which an agent perceives and acts upon the world using textual natural language. These games are usually structured as puzzles or quests with long-term dependencies in which an agent must complete a sequence of actions to succeed—providing ideal environments in which to test an agent's ability to explain its actions. Our agent is designed to treat explainability as a first-class citizen, using an extracted symbolic knowledge graph-based (KG) state representation coupled with a Hierarchical Graph Attention mechanism that points to the facts in the internal graph representation that most influenced the choice of actions. Experiments show that this agent provides significantly improved explanations over strong baselines, as rated by human participants generally unfamiliar with the environment, while also matching state-of-the-art task performance.

## 1 Introduction

Explainable AI refers to artificial intelligence methods and techniques that provide human-understandable insights into how and why an AI system chooses actions or makes predictions. Such explanations are critical for ensuring reliability and improving trustworthiness by increasing user understanding of the underlying model. In this work we specifically focus on creating deep reinforcement learning (RL) agents that can explain their actions in sequential decision making environments through natural language.

In contrast to the majority of contemporary work in the area which focuses on supervised machine learning problems which require singular instance level *local explanations* (You et al., 2016; Xu et al., 2015; Wang et al., 2017; Wiegreffe and Marasovic, 2021), such environments—in which agents need to reason causally about actions over a long series

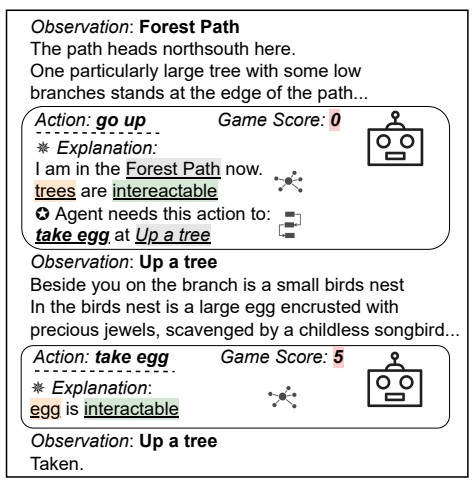

Figure 1: Excerpt from *zork1* with *immediate step-by-step explanations* constructed from the KG represented by ⚙ and *temporally extended explanations* by ⊟ . Colors represent different categories of KG facts seen in Fig. 2.

---

[1]Code: https://github.com/xiangyu-peng/HEX-RL

36th Conference on Neural Information Processing Systems (NeurIPS 2022).

of steps—require an agent to take into account both environmentally grounded context as well as goals when producing explanations. Agents implicitly contain beliefs regarding the downstream effects—the changes to the world—that actions taken at the current timestep will have. This requires explanations in these environments to contain an additional *temporally extended* component taking the full trajectory's context into account—complementary to the *immediate* step-by-step explanations.

Interactive Fiction (IF) games (Fig. 1) are partially observable environments where an agent perceives and acts upon a world using potentially incomplete textual natural language descriptions. They are structured as long puzzles and quests that require agents to reason about thousands of locations, characters, and objects over hundreds of steps, creating chains of dependencies that an agent must fulfill to complete the overall task. They provide ideal experimental test-beds for creating agents that can both reason in text and explain it.

We introduce an approach to game playing agents—**Hierarchically Explainable Reinforcement Learning (HEX-RL)**—that is designed to be *inherently* explainable, in the sense that its internal state representation—i.e. belief state about the world—takes the form of a symbolic, human-interpretable knowledge graph (KG) that is built as the agent explores the world. The graph is encoded by a Graph Attention network (GAT) (Veličković et al., 2017) *extended* to contain a hierarchical graph attention mechanism that focuses on different sub-graphs in the overall KG representation. Each of these sub-graphs contains different information such as attributes of objects, objects the player has, objects in the room, current location, etc. Using these encoding networks in conjunction with the underlying world KG, the agent is able to create *immediate explanations* akin to a running commentary that points to the facts within this knowledge graph that most influence its current choice of actions when attempting to achieve the tasks in the game on a step-by-step basis.

While graph attention can tell us which elements in the KG are attended to when maximizing expected reward from the current state, it cannot explain the intermediate, unrewarded dependencies that need to be satisfied to meet the long term task goals. For example, in the game *zork1,* the agent needs to pick up a lamp early on in the game—an unrewarded action—but the lamp is only used much later on to progress through a location without light. Thus, our agent additionally analyzes an overall episode trajectory—a sequence of knowledge graph states and actions from when the agent first starts in a world to either task completion or agent death—to find the intermediate set of states that are most important for completing the overall task. This information is used to generate a *temporally extended explanation* that condenses the *immediate step-by-step explanations* to only the most important steps required to fulfill dependencies for the task.

Our contributions are as twofold: (1) we create an inherently explainable agent that uses an ever-updating knowledge-graph based state representation to generate step-by-step immediate explanations for executed actions as well as performing a post-hoc analysis to create temporal explanations; and (2) a thorough experimental study against strong baselines that shows that our agent generates significantly improved explanations for its actions when rated by human participants *unfamiliar with the domain* while not losing any task performance compared to the current state-of-the-art knowledge graph-based agents.

## 2    Background and Related Work

Interactive Fiction (IF) games are simulations featuring language-based state and action spaces. It provides a platform for exploring lifelong open-domain dialogue learning (Shuster et al., 2020) and action elimination with deep reinforcement learning (Zahavy et al., 2018). In this paper, we use IF games as our test-bed because they provide an ideal platform for collecting data, linking game states and actions to the corresponding natural language explanations. We use the definition of text-adventure games as seen in Côté et al. (2018) and Hausknecht et al. (2020). We take Jericho (Hausknecht et al., 2020), a framework for interacting with text games, as the interface connecting learning agents with interactive fiction games. A text game can be defined as a partially-observable Markov Decision Process: $G = \langle S, P, A, O, \Omega, R, \gamma \rangle$, representing the set of environment states, conditional transition probabilities between states, the vocabulary or words used to compose text

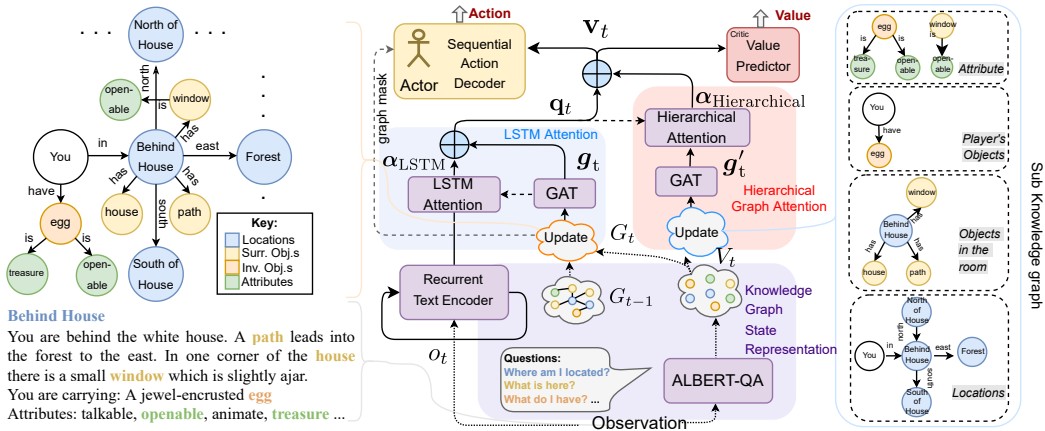

Figure 2: Knowledge graph extraction and the (HEX-RL) agent's architecture at step $t$.

commands, observations, observation conditional probabilities, reward function, and discount factor, respectively. The reinforcement learning agent is trained to learned a policy $\pi_G(o) \rightarrow a$.

**Knowledge Graphs for Text Games.** Ammanabrolu et al. (2020) proposed Q*BERT, a reinforcement learning agent that learns a KG of the world by answering questions. Xu et al. (2020) introduce the SHA-KG, a stacked Hierarchical Graph Attention mechanism to construct an explicit representation of the reasoning process by exploiting the structure of the KG. Adhikari et al. (2020) present the Graph-Aided Transformer Agent (GATA) which learns to construct a KG during game play and improves zero-shot generalization on procedurally generated TextWorld games. Other works such as Murugesan et al. (2020) explore how to use KGs to endow agents with commonsense. While these works showcase the effectiveness of KGs on task performance and do not focus on how explainable their architectures are. We further note that these architectures do now allow for as fine-grained attention-based attribution as HEX-RL's architecture does—e.g. Q*BERT does not use relationship information in their policy and SHA-KG averages attention across large portions of the graph and is unable to point to specific triples in its KG representation to explain an action.

**Explainable Deep RL.** Contemporary work on explaining deep reinforcement learning policies can be broadly categorized based on: (1) how the information is extracted, either via intrinsic motivation during training (Shu et al., 2017; Hein et al., 2017; Verma et al., 2018) or through post-hoc analysis (Rusu et al., 2015; Hayes and Shah, 2017; Juozapaitis et al., 2019; Madumal et al., 2020); and (2) the scope—either global (Zahavy et al., 2016; Hein et al., 2017; Verma et al., 2018; Liu et al., 2018) or local (Shu et al., 2017; Liu et al., 2018; Madumal et al., 2020; Guo et al., 2021). In our work, we create an agent that spans more than one of these categories providing immediately local explanations through extracted knowledge graph representations and post-hoc temporal explanations. Inspired by Madumal et al. (2020), we learn a graphical causal model which focuses on using relations between steps in a puzzle to generate temporal explanations instead of generating counterfactuals.

## 3 Hierarchically Explainable RL

Our work aims to generate (1) *immediate step-by-step explanations* of an agent's policy by capturing the importance of the current game state observation and (2) *temporally extended explanations* that take into context an entire trajectory via a post-hoc analysis. Formally, let $\mathbf{X} = \{\mathbf{s}_t, \mathbf{a}_t\}_{t=1:T}$ be the set of game steps that compose a trajectory. Each game state $s_t$ consists of a knowledge graph $G_t$ representing all the information learned since the start of the game. This graph is further split into four sub-knowledge graphs each containing different, semantically related relationship types. This section first describes a graph attention based architecture that uses these sub-graphs to produce immediate explanations. We then describe how to filter the game states in a trajectory into a condensed set of the most important ones $\mathbf{X}' \subset \mathbf{X}$ that best capture the underlying dependencies that need to be fulfilled to complete the task—enabling us to produce temporal explanations.

**Knowledge Graph State Representation.** Building on Ammanabrolu et al. (2020), constructing the knowledge graph is treated as a question-answering task. KGs in these games take the form of RDF triples (Klyne, 2004) of $\langle subject, relation, object \rangle$—extracted from text observations and update as the agent explores the world. The agent answers questions about the environment such as, "What am I carrying?" or "What objects are around me?". A specially constructed dataset for question answering in text games—JerichoQA—is used to fine-tune ALBERT (Lan et al., 2019) to answer these questions (See Appendix A.3). The answers form a set of candidate graph vertices $V_t$ for the current step and questions form the set of relations $R_t$. Both $V_t$ and $R_t$ are then combined with the graph at the previous step $G_{t-1}$ to update the agent's belief about the world state into $G_t$. The left side of Figure 2 showcases this.

In an attempt to enable more fine grained explanation generation and inspired by Xu et al. (2020), we divide the knowledge graph $G$ into multiple sub-graphs $G^{atr}, G^{inv}, G^{obj}, G^{loc}$, each representing (1) attributes of objects, (2) objects the player has, (3) objects in the room, and (4) other information such as location (right side of Fig. 2) based on the corresponding relationship types extracted by the ALBERT-QA module. The union of all sub-graphs is equivalent of $V_t$ and $R_t$ extracted from the current game state. The full knowledge graph $G_t$ captures the overall game state since the start of the game. The sub-graphs easily reflect different relationships of the current game state.

**Template Action Space.** Agents output a language string into the game to describe the actions that they want to perform. To ensure tractability, this action space can be simplified down into templates. Templates consist of interchangeable verbs phrases ($VP$), optionally followed by prepositional phrases ($VP\ PP$), e.g. ($[carry/take]$ __) and ($[throw/discard/put]$ __ $[against/on/down]$ __), where the verbs and prepositions within [.] are aliases. Actions are constructed from templates by filling in the template's blanks using words in the game's vocabulary. Size of action space is shown in Appendix A.1.

## 3.1 Immediate Explanations

Our immediate explanations consist of finding the subset of triplets in sub-graphs $G^{atr}, G^{inv}, G^{obj}, G^{loc}$ c the action decision made at the current step—is capable of explaining the action. We introduce a deep RL architecture capable of this.

**Hierarchical Knowledge Graph Attention Architecture.** At each step, a total score $R_t$ and an observation $o_t$ is received—consisting of $\left(o_{t_{\text{desc}}}, o_{t_{\text{game}}}, o_{t_{\text{inv}}}, a_{t-1}\right)$ corresponding to the room description, game feedback, inventory, and previous action and are processed using a GRU based encoder using the hidden state from the previous step, combining them into a single observation embedding $\mathbf{o}_t \in \mathbb{R}^{d_{text} \times c}$ (bottom of Fig. 2).

The full knowledge graph $G_t$ is processed via Graph Attention Networks (GATs) (Veličković et al., 2017) followed by a linear layer to get the graph representation $\mathbf{g_t} \in \mathbb{R}^{d_{text}}$ (middle of Fig. 2). We compute *LSTM attention* between $\mathbf{o}_t$ and $\mathbf{g_t}$ as:

$$\boldsymbol{\alpha}_{\text{LSTM}} = \text{softmax}\left(\boldsymbol{W}_l \boldsymbol{h}_{\text{LSTM}} + \boldsymbol{b}_l\right) \tag{1}$$

$$\boldsymbol{h}_{\text{LSTM}} = \tanh\left(\boldsymbol{W}_o \boldsymbol{o}_t \oplus \left(\boldsymbol{W}_g \mathbf{g_t} + \boldsymbol{b}_g\right)\right) \tag{2}$$

where $\oplus$ denotes the addition of a matrix and a vector. $\boldsymbol{W}_l \in \mathbb{R}^{d_{text} \times d_{text}}$, $\boldsymbol{W}_g \in \mathbb{R}^{d_{text} \times d_{text}}$, $\boldsymbol{W}_o \in \mathbb{R}^{d_{text} \times d_{text}}$ are weights and $\boldsymbol{b}_l \in \mathbb{R}^{d_{text}}$, $\boldsymbol{b}_o \in \mathbb{R}^{d_{text}}$ are biases. The overall representation vector is updated as:

$$\mathbf{q_t} = \mathbf{g_t} + \sum_i^c \boldsymbol{\alpha}_{\text{LSTM},i} \odot \boldsymbol{o}_{\text{t},i} \tag{3}$$

where $\odot$ denotes dot-product and $c$ is the number of $\boldsymbol{o}_{\text{t}}$'s components.

Sub-graphs are also encoded by GATs to get the graph representation $\mathbf{g}'_{\mathbf{t}} \in \mathbb{R}^{d_{G_t,sub} \times m}$ (no. of subgraphs). The *Hierarchical Graph Attention* between $\mathbf{q}_t \in \mathbb{R}^{d_{G_t,sub}}$[2] and $\mathbf{g}'_{\mathbf{t}}$ is calculated by:

$$\boldsymbol{\alpha}_{\text{Hierarchical}} = \text{softmax}\left(W_{\text{H}} \boldsymbol{h}_{\text{H}} + \boldsymbol{b}_{\text{H}}\right) \tag{4}$$

$$\boldsymbol{h}_{\text{H}} = \tanh\left(W_{\text{g}'} \boldsymbol{g}'_{\text{t}} \oplus \left(W_{\text{q}} \boldsymbol{q}_{\text{t}} + \boldsymbol{b}_{\text{q}}\right)\right) \tag{5}$$

where $W_{\text{H}} \in \mathbb{R}^{d_{G_t,sub} \times d_{G_t,sub}}$, $W_{\text{g}'} \in \mathbb{R}^{d_{G_t,sub} \times d_{G_t,sub}}$, $W_{\text{q}} \in \mathbb{R}^{d_{G_t,sub} \times d_{G_t,sub}}$ are weights and $\boldsymbol{b}_{\text{H}} \in \mathbb{R}^{d_{G_t,sub}}$, $\boldsymbol{b}_{\text{q}} \in \mathbb{R}^{d_{G_t,sub}}$ are biases. Then we get state representation, consisting of the textual observations full knowledge graph and sub-knowledge graph.

$$\mathbf{v_t} = \mathbf{q_t} + \sum_{i}^{s} \boldsymbol{\alpha}_{\text{Hierarchical},i} \odot \boldsymbol{g}'_{\text{t},i} \tag{6}$$

where $s$ is the number of sub-graphs (4 in our paper). The full architecture can be found in Figure 2.

The agent is trained via the Advantage Actor Critic (A2C) (Mnih et al., 2016) method to maximize long term expected reward in the game in a manner otherwise unchanged from Ammanabrolu et al. (2020) (See Appendix A.2). These attention values thus reflect the portions of the knowledge graphs that the agent must focus on to best achieve this goal of maximizing reward.

**Hierarchical Graph Attention Explanation.** The graph attention $\boldsymbol{\alpha}_{\text{Hierarchical}}$ is used to capture the relative importance of game state observations and KG entities in influencing action choice. For each sub-graph, the graph attention, $\boldsymbol{\alpha}_{\text{Hierarchical},i} \in \mathbb{R}^{n_{\text{nodes}} \times m}$ is summed over all the channels $m$ to obtain $\boldsymbol{\alpha}'_{\text{Hierarchical},i} \in \mathbb{R}^{n_{\text{nodes}} \times 1}$, showing the importance of the KG nodes in the $i$th sub-graph. The top-$k$ valid entities (and corresponding edges) with highest absolute value of its attention form the set of knowledge graph triplets that best locally explain the action $a_t$.

In order to make the explanation more readable for a human reader, we further transform knowledge graph triplets to natural language by template filling.

We create templates for each type of sub-graphs $G^{atr}, G^{inv}, G^{obj}, G^{loc}$.

- $\langle object, is, attribute \rangle \rightarrow$ "*Object is attribute*"
- $\langle player, has, object \rangle \rightarrow$ "*I have object*"
- $\langle object, in, location \rangle \rightarrow$ "*Object is in location*"
- $\langle location1, direction, location2 \rangle \rightarrow$ "*location 1 is in the direction of location 2*", e.g. $\langle forest, north, house \rangle$ is converted to "*Forest is in the north of house*"

More examples can be found in Appendix A.4.

## 3.2 Temporally Extended Explanations

Graph attention tells us which entities in the KG are attended to when making a decision, but is not enough alone for explaining "why" actions are the right ones in the context of fulfilling dependencies that may potentially be unrewarded by the game—especially given the fact that there are potentially multiple ways of achieving the overall task. HEX-RL thus saves trajectories for hundreds of test time rollouts of the games, performed once a policy has been trained (Table 1 and Appendix A.5). The game trajectories consist of all the game states, actions taken, predicted critic values, game scores, the knowledge graphs, and the immediate step level explanations generated as previously described. HEX-RL produces a temporal explanation by performing a post-hoc analysis on these game trajectories. The agent then analyzes and filters these trajectories in an attempt to find the subset of states that are most crucial to achieving the task as summarized in Figure 3—then using that subset of states to generate temporal trajectory level explanations.

**Bayesian State Filter.** We first train a Bayesian model to predict the conditional probability $\mathbb{P}(A \mid B_i)$ of a game step $(A)$ given any other possible game step $(B_i)$ in the game trajectories. More specifically, each game step is composed of 3 elements, game state $o_t$, action $a_t$ and the current knowledge graph $G_t$. The key intuition here being that state, action pairs that appear in a certain ordering in multiple trajectories are more likely to dependant on each other.

---

[2]A linear transformation ensures that $\mathbf{q}_t \in \mathbb{R}^{d_{G_t,sub}}$.

The set of game steps with the highest $\mathbb{P}(A \mid B_i)$ is used to explain taking the action associated with game state $A$. For example, "take egg" ($A$) is required to "open egg" ($B$), and $\mathbb{P}(A \mid B) = 1$, hence "open egg" is used as a reason why action "take egg" must be taken first. The initial set of game states $\mathbf{X}$ is filtered into $\mathbf{X}_1$ by working backwards from the final goal state by finding the set of states that form the most likely chain of causal dependencies that lead to it. Details can be found in Appendix A.5 and A.6.

**Language Model Action Filter.** Following this, we apply a GPT-2 (Radford et al., 2019) language model trained to generate actions based on transcripts of text games from human play-throughs to further filter out important states—known as the Contextual Action Language Model (CALM) (Yao et al., 2020). As this language model is trained on human transcripts, we hypothesize that it is able to further filter down the set of important states by finding the states that have corresponding actions that a human player would be more likely to perform—thus potentially leading to more natural explanations. CALM takes into observation $o_t$, action $a_t$ and the following observation $o_{t+1}$, and predicts next valid actions $a_{t+1}$. In our work, we use CALM as a filter to look for the relations between a game step $A$ and the explanation candidates $B_i \in \mathbf{X}_1$. We feed CALM with the prompt $o_A, a_A, o_{B_i}$ to get an action candidate set. When the two game steps $A$ and $B_i$ are highly correlated, given $o_A$, $a_A$ and $o_{B_i}$, CALM should successfully predict $a_{B_i}$

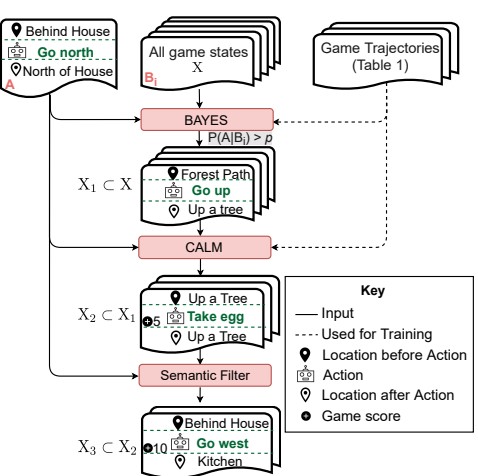

Figure 3: Temporal explanation pipeline for why the agent chose the action—"*go north*" at "*Behind House*".

with high probability. The game steps $B_i$, whose associated action $a_{B_i}$ is in this generated action candidates set, are saved as the next set of filtered important candidate game states ($\mathbf{X}_2$).

**Semantic State-Action Filter.** To better account for the irregularities of the puzzle like environment, we adopt a *semantic filter* to obtain the final important state set $\mathbf{X}_3$. Here, given $A, B_i \in \mathbf{X}_2$, states are further filtered on the basis of whether one of these scenarios occurs: (1) $a_A$ and $a_{B_i}$ contain the same entities, e.g. "*take egg*" and "*open egg*". (2) $G_A$ and $G_{B_i}$ share the same entities, e.g. "lamp" occurs in both observations. (3) $A$ and $B_i$ occur in the same location, e.g. after taking action $a_A$, the player enters "kitchen" and $B$ occurs in "kitchen". (4) The state has a non-zero reward or a high absolute critic value, indicating that

Table 1: Example state saved during game play.

| |
|---|
| STEP: 16 |
| **Text Observation:**
Up a Tree
Beside you on the branch is a small birds nest.
In the birds nest is a large egg encrusted with jewels...
**Knowledge graph:**
$\langle tree, in, forest \rangle, \langle egg, is, interactable \rangle...$ |
| **Action:** take egg
**Immediate explanation**: egg is interactable |
| **Game Score:**5
**Critic Value**: 5.7457 |

it is either a state important for achieving the goals of the game or it is a state to be avoided. The final set of important game states $\mathbf{X}_3$ is used to synthesize post-hoc temporal explanations for why an action was performed in a particular state—as seen in Figure 1—taking into account the overall context of the dependencies required to be satisfied and building on the immediate step level explanations for each given state in $\mathbf{X}_3$. Ablation studies pin-pointing the relative contributions of the different filters are found in Section 4.4. We concluded that all three steps of the filtering process to identify important states are necessary for creating coherent temporal explanations that effectively take into account the context of the agent's goals.

Table 2: Asymptotic scores on games by different methods across 5 independent runs. *Eps.* indicates normalized scores averaged across 100 episodes of testing which occurs at the end of the training period and *Max* indicates the maximum score seen by the agent over the same period. We present results on two training rewards for HEX-RL, *game only* and *game and IM*.

| Experiment | LSTM-A2C | | KG-A2C | | SHA-KG | | Q*BERT | | HEX-RL *Game Only* | | HEX-RL *Game and IM* | | Max |
|---|---|---|---|---|---|---|---|---|---|---|---|---|---|
| Metric | Eps. | Max | Eps. | Max | Eps. | Max | Eps. | Max | Eps. | Max | Eps. | Max | - |
| zork1 | 27 | 31.2 | 34 | 35 | 33.6 | 34.5 | 35 | 35 | 29.8 | 40 | 30.2 | 40 | 350 |
| library | 8.2 | 10 | 14.3 | 19 | 10.0 | 15.8 | 18 | 18 | 16.0 | 19 | 13.8 | 21 | 30 |
| detective | 141 | 188 | 207.9 | 214 | 246.1 | 308 | 274 | 310 | 276.7 | 330 | 276.9 | 330 | 360 |
| balances | 10 | 10 | 10 | 10 | 9.8 | 10 | 10 | 10 | 10.0 | 10 | 10.0 | 10 | 51 |
| pentari | 50.4 | 55 | 50.7 | 56 | 48.2 | 51.3 | 50 | 56 | 34.6 | 55 | 44.7 | 60 | 70 |
| ztuu | 5 | 5 | 5 | 5 | 5 | 25 | 5 | 5 | 5.0 | 5 | 5.1 | 9 | 100 |
| ludicorp | 14.4 | 18 | 17.8 | 19 | 17.6 | 17.8 | 18 | 19 | 14.0 | 18 | 17.6 | 18 | 150 |
| deephome | 1 | 1 | 1 | 1 | 1 | 1 | 1 | 1 | 1.0 | 1 | 1.0 | 1 | 300 |
| temple | 8 | 8 | 7.6 | 8 | 7.9 | 6.9 | 8 | 8 | 8.0 | 8 | 7.6 | 8 | 35 |
| % compl. | 22.6 | 25.9 | 27.3 | 30.8 | 27.2 | 33.1 | **30.8** | 34.9 | 27.2 | 33.9 | 28.2 | **35.8** | 100 |
| std dev | 0.02 | 0.01 | 0.06 | 0.01 | - | - | 0.03 | 0.00 | 0.03 | 0.01 | 0.03 | 0.02 | - |

## 4 Evaluation

Our evaluation consists of three phases: (1) We show that HEX-RL has the comparable performance to state-of-art reinforcement learning agents on text games in Section 4.1. (2) Then in Section 4.2, we evaluate our immediate attention explanation model by comparing the explanations generated by HEX-RL and agents that do not use knowledge graphs (See Fig. 2 and Section 3.1). (3) In Section 4.3 we compare immediate to temporal explanations, focusing on the effects that including trajectory level context when evaluating explanations in the context of agent goals. (4) In Section 4.4 we conduct human participant ablation study evaluating the individual contributions of the filtration pipeline for generating temporal explanations seen in Figure 3.

### 4.1 Task Performance Evaluation

We compare HEX-RL with four strong state-of-art reinforcement learning agents—focusing on contemporary agents that use knowledge graphs—on an established test set of 9 games from the Jericho benchmark (Hausknecht et al., 2020).

- **LSTM-A2C** is a baseline that only uses natural language observations as state representation that is encoded with an LSTM-based policy network.

- **KG-A2C.** Instead of training a question-answering system like Q*BERT to build knowledge graph state representation, KG-A2C (Ammanabrolu and Hausknecht, 2020) extracts knowledge graph triplets from the text observations using a rules based approach built on OpenIE (Angeli et al., 2015).

- **SHA-KG** is adapted from Xu et al. (2020) and uses a rules-based approach to construct a knowledge graph for the agent which is then fed into a Hierarchical Graph Attention network as in HEX-RL. This agent separates the sub-graphs out using a rules-based approach and makes no use of any graph edge relationship information.

- **Q*BERT.** Ammanabrolu et al. (2020) uses a similar method of creating the knowledge graph through question answering but does not use the hierarchical graph attention architecture combined with the sub-graphs.

These baselines are all trained via the Advantage Actor Critic (A2C) (Mnih et al., 2016) method—further comparisons to other contemporary agents can be found in Appendix A.7. It is also worth noting that most contemporary state of the art deep RL agents for text games use recurrent neural policy networks as opposed to transformer networks due to their improved performance in this domain.

**HEX-RL Training.** We trained HEX-RL on two reward types: (a) *game only*, which indicates that we only use score obtained from the game as reward. (2) *game with intrinsic motivation (game*

*and IM)*, which contains an additional intrinsic motivation reward based on knowledge graph expansion as seen in Ammanabrolu et al. (2020)—where the agent is additionally rewarded for learning more about the world by finding new facts for knowledge graph (see Appendix A.8, A.9 and A.10).

Table 2 shows the performance of HEX-RL and the other four baselines. We can see that designing the HEX-RL agent to be inherently explainable through the use of Hierarchical Graph Attention and the sub-graphs improves the overall maximum score seen during training when compared to any of the other agents. In terms of the average score seen during the final 100 episodes, HEX-RL wth intrinsic motivation outperforms all baselines with the exception of Q*BERT—there HEX-RL significantly outperforms Q*BERT on one game, is outperformed on two games, and comparable on the remaining six games. HEX-RL thus performs comparably to other state-of-the-art baselines in terms of overall task performance while also boasting the additional ability to explain its actions.

## 4.2 Immediate Explanation Evaluation

Having established that HEX-RL's performance while playing text games is comparable to other state-of-the-art agents, we attempt to answer the question of exactly how useful the knowledge graph based architecture is when generating immediate step-by-step explanations by comparing HEX-RL to a baseline that doesn't use knowledge graphs in a human participant study. Two models for step-by-step explanations are compared:

- **LSTM Attention explanations.** Extracts the most important substring in the observations through LSTM attention $\alpha_{LSTM}$ and then uses those words to create an explanation.
- **Hierarchical Graph Attention explanations.** Extracts KG triplets most influenced the choice of actions by Hierarchical Attention $\alpha_{\text{Hierarchical}}$ and then transforming them into readable language explanations through templates.

We recruited 40 participants—generally unfamiliar with the environment at hand—on a crowd sourcing platform. Each participant reads a randomly selected subset of 10 explanation pairs (drawn randomly from a pool totaling 60 explanation pairs), generated by Hierarchical Graph Attention and LSTM attention explanation on three games in the Jericho benchmark: *zork1*, *library*, and *balances*. We choose three games with very different structures and genres as defined in Hausknecht et al. (2020). They each require a diverse set of action types and solutions to complete and thus provide a wide area of coverage when used as test beds for human evaluation of explanations. Then they are given the following metrics and asked to choose which explanation they prefer for that metric:

- *Confidence*: This explanation makes you more confident that the agent made the right choice.
- *Human-likeness*: This explanation expresses more human-like thinking on the action choice.
- *Understandability*: This explanation makes you understand why the agent made the choice.

Variations of these questions have been used to evaluate other explainable AI systems (eg. Ehsan et al. (2019)). At least 5 participants give their preference for each explanation pair. We take the majority preference from humans participants as the result. More details are shown in Appendix B.1.

Figure 4 shows the result of the human evaluation of attention explanations. Hierarchical graph attention explanation is preferred over LSTM attention explanation in all three dimensions. These results are statistically significant ($p < 0.05$) with fair inter-rater reliabilities. We also observe that these three dimensions are highly, positively correlated using Spearman's Rank Order Correlation.[3]

A slightly higher proportion of participants preferred the LSTM Attention explanations in the human-likeness dimension compared to the other two. The participants preferring LSTM Attention explanation stated that they found it intuitive but often incoherent and found the Hierarchical Graph Attention explanations to be more robotic. LSTM attention explanations are substrings of the human-written observation and thus have the potential to be more natural sounding than

---

[3]$r_s = 0.70$, $p < 0.01$, between "confidence" and "understandability"; $r_s = 0.67$, $p < 0.01$, between "confidence" and "human-likeness"; $r_s = 0.89$, $p < 0.01$, between "human-likeness" and "understandability"

the templated Hierarchical Graph Attention explanations when they are coherent enough to be understood. The KG sacrifices a small amount of human-likeness in return for much greater overall coherence and accuracy. KGs with Hierarchical Graph Attention give us explanations that are more easily understood and inspire greater confidence in the agent's decisions.

Example qualitative LSTM Attention and Hierarchical Graph Attention explanations can be found in Appendix C.2. As our system relies on graph hierarchical graph attention to generate immediate explanations, a well-trained knowledge graph representation module of the world knowledge is required. Most cases where the agent fails to provide satisfactory immediate explanations are either when: (1) the explanation is not directly linked to the one of the facts we choose to extract from the knowledge graph, such as object/location information; and (2) due to the error of knowledge graph extraction models themselves.



Figure 4: Human evaluation results showing the proportion of participants that prefer Hierarchical Graph Attention vs. LSTM Attention explanations, $**$ indicates $p < 0.01$, † indicates $\kappa > 0.2$ or fair agreement. ‡ indicates $\kappa > 0.4$ or moderate agreement. Evaluation results on each game are shown in Appendix A.11.

### 4.3  Immediate vs. Temporal Explanations

Having proved the effectiveness of the knowledge graph at the immediate step-by-step explanation level, we evaluate our method of producing temporal explanations and how they compare to the immediate explanations along two dimensions: (1) coherence; and (2) explanation accuracy when taken in the context of the agent's goals.

Participants first read a trajectory of the game combined with step-by-step immediate explanations and the game goal, then indicate how much they agree with the statements on a Likert scale of 1 (strong disagree) to 5 (strong agree). Here, we add two metrics from the previous study:

- *Goal context*: You are able to understand why the agent takes this particular sequence of actions given what you know about the goal.
- *Readability*: This explanation is easy to read.

Figure 5 shows the average scores for each question for the immediate and temporal explanations. The temporal explanations achieve comparable performance to the immediate explanations on all metrics except for the the metric relating to goal context where they significantly out-perform the immediate explanations. A majority of participants stated that a condensed trajectory level explanation made the goals of the agent easier to understand than reading through each step level explanation. These results indicate that HEX-RL can generally successfully identify the most important states in a trajectory and use them to create temporal explanations that are on par with immediate explanations in terms of coherence but provide significantly more context

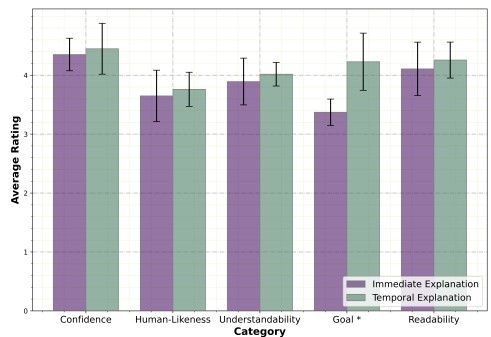

Figure 5: Human judgment[5] results on immediate and temporal explanation, $*$ indicates $p < 0.05$. Error bars indicate a confidence level of 95%.

in terms of explaining an agent's actions with respect to its task-based goals.

Cases where the agent does not provide a temporally coherent and goal-driven explanation revolve around failures—particularly by the Bayesian State Filter—in detecting the most important states in the trajectory. A qualitative analysis (as seen in Appendix C.3) suggests that this occurs in cases where there are a large number of branching paths that lead to the same end state. Thus, the quality of the generated temporal explanations appears to be inversely proportional to the relative complexity of the game as measured by its branching factor.

### 4.4 Temporal Explanation Ablation Study

Having established the overall effectiveness of the filters in HEX-RL that create the temporal explanations, we perform pair-wise ablation studies to pinpoint the relative contributions of the different filters seen in Fig. 3. We first compare explanations generated using a set of important states filtered from the trajectory using the Bayes model compared to Bayes+CALM explanation. This how applying the language model action filter affects the quality of the temporal explanations. As before, we recruited 30 participants on a crowd sourcing platform. Each participant reads a randomly selected subset of explanation pairs, comprised of temporal explanations filtered by Bayes and Bayes+CALM models. Figure 6a shows that after applying the CALM model to filter explanation candidates, generated explanations are significantly preferred on the "Confidence" and "Understandability" dimensions.

Similarly, we then conducted another ablation study to validate the contribution of semantic filter by comparing the Bayes+CALM filtering method to the full HEX-RL using Bayes+CALM+Semantic filters. The experiment setup is the same as the previous ablation study. Figure 6b shows that Bayes+CALM+Semantic performs significantly better than Bayes+CALM on all three dimensions.

We additionally observe that these three metrics are highly, positively correlated using Spearman's Rank Order Correlation in both of these ablation studies[4]. When asked to justify their choices, participants indicated that the full HEX-RL system with Bayes+CALM+Semantic filters provided temporal explanations that they felt was more understandable than alternatives. These results indicate that all three steps of the filtering process to identify important states are necessary for creating coherent temporal explanations that effectively take into account the context of the agent's goals.



(a) Bayes vs. Bayes + CALM explanation



(b) Bayes + CALM vs. Bayes + CALM + Semantic explanation

Figure 6: Human evaluation results on ablation study, $*$ indicates $p < 0.05$, $\dagger$ indicates $\kappa > 0.2$ or fair agreement.

## 5 Conclusions

Explaining deep RL policies for sequential decision making problems in natural language is a sparsely studied problem despite a steadily growing need. An oft given reason for this phenomenon is that deep RL methods perform better without the additional burden of being explainable. In an attempt to encourage work in this area, we create the Hierarchically Explainable Reinforcement Learning (HEX-RL) agent which treats explainability as a first-class citizen in its design by using a readily interpretable knowledge graph state representation coupled with a Hierarchical Graph Attention network. This agent is able to produce step-by-step commentary-like immediate explanations and also a condensed temporal trajectory level explanation via a post-hoc analysis. We show that with careful design, it is possible to create inherently explainable RL agents that do not lose performance when compared to contemporary state-of-the-art agents and simultaneously are able to generate significantly higher quality explanations of actions.

---

[4]$r_s = 0.86$, $p < 0.01$, between "confidence" and "understandability"; $r_s = 0.79$, $p < 0.01$, between "confidence" and "human-likeness"; $r_s = 0.90$, $p < 0.01$, between "human-likeness" and "understandability"

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
