# A  Implementation Details

## A.1  Full Action Space Size

Jericho provides the capability to extract game-specific vocabulary and action templates (Section 3). These templates contain up to two blanks, so a typical game with 200 templates and a 700 word vocabulary yields an action space of $\mathcal{O}(TV^2) \approx 98$ million, three orders of magnitude smaller than the 240-billion space of 4-word actions using vocabulary alone.

## A.2  A2C Architecture

Further details of what is found in Figure 2. The sequential action decoder consists two GRUs that are linked together as seen in Ammanabrolu and Hausknecht (2020). The first GRU decodes an action template and the second decodes objects that can be filled into the template. These objects are constrained by a *graph mask*, i.e. the decoder is only allowed to select entities that are already present in the knowledge graph.

Same with Ammanabrolu and Hausknecht (2020), the loss consists of template loss, object loss, value loss, actor loss and entropy loss. The template loss given a particular state and current network parameters is applied to the decoder. Similarly, the object loss is applied across the decoder is calculated by summing cross-entropy loss from all the object decoding steps. Entropy loss over the valid actions, is designed to prevent the agent from prematurely converging on a trajectory. The following hyperparameters are taken from the original paper and known to work well on text games.

| Parameters | Value |
|---|---|
| discount factor | 0.9 |
| entropy coefficient | 0.03 |
| value coefficient | 9 |
| template coefficient | 3 |
| object coefficient | 9 |

## A.3  Knowledge Graph Representation QA Model

The question answering network based on ALBERT (Lan et al., 2019) has the following hyperparameters, taken from the original paper and known to work well on the SQuAD 2.0 (Rajpurkar et al., 2018) dataset. No further hyperparameter tuning was conducted.

| Parameters | Value |
|---|---|
| batch size | 8 |
| learning rate | 3e-5 |
| max seq len | 512 |
| doc stride | 128 |
| warmup steps | 814 |
| max steps | 8144 |
| gradient accumulation steps | 24 |

## A.4  Templates of Immediate Explanation

We consider four types of sub-graphs $G^{atr}, G^{inv}, G^{obj}, G^{loc}$, each representing (1) attributes of objects, (2) objects the player has, (3) objects in the room, and (4) other information such as location (see right side of Figure 2). Hence, we create one template for each sub-graph with converting conjugated forms of verbs,

- $\langle object, is, attribute \rangle$ is converted to "*Object is attribute*".
  For example, $\langle trees, is, interactable \rangle$ is converted to "*trees are interactable.*". $\langle egg, is, interactable \rangle$ is converted to "*egg is interactable*".
- $\langle player, has, object \rangle$ is converted to "*I have object*".
  For example, $\langle player, has, eggs \rangle$ is converted to "*I have eggs*". $\langle player, has, knife \rangle$ is converted to "*I have knife*".

- $\langle object, in, location \rangle$ is converted to "*Object is in location*".
  For example, $\langle egg, in, forest \rangle$ is converted to "*egg is in forest*".
  $\langle trees, in, forest \rangle$ is converted to "*trees are in forest*".
- $\langle location\_1, direction, location\_2 \rangle$ is converted to "*location_1 is in the direction of location_2*".
  For example, $\langle forest, north, house \rangle$ is converted to "*forest is in the north of house*".

## A.5 HEX-RL explanation parameters

There are some other parameters which may affect the results of explanations. Our initial experiments suggested that the larger the number of trajectories we use, the more accurate the Bayesian State Filter.

| Parameter | Value | Explanation of Use |
|---|---|---|
| trajectories for test time rollouts | 300 | Number of saved trajectories of test time rollouts of the games, which is performed once a policy has been trained |
| Bayesian State Filter threshold | 0.5 | Larger threshold filters more steps out. |
| topk of CALM | 20 | Topk sampling of CALM model |

## A.6 Bayesian State Filter Detail

We first train a Bayesian model to predict the conditional probability $\mathbb{P}(A \mid B_i)$ of a game step ($A$) given any other possible game step ($B_i$) in the game trajectories. More specifically, current game step ($A$) is composed of 3 elements, game state $o_t$, action $a_t$ and knowledge graph $G_t$. We count the occurrence of $A$ ($\mathbb{C}(A)$), and all the game steps occurred in the game logs ($\mathbb{C}(B_i)$), and also count the co-occurrence of $A$ and $B_i$, $\mathbb{C}(A \cap B_i)$ in the same trajectory. $\mathbf{X} = \{A, B_1, ..., B_i\}$. The conditional probability $\mathbb{P}(A \mid B_i)$ is calculated by,

$$\mathbb{P}(A \mid B_i) = \frac{\mathbb{P}(B_i \mid A)\mathbb{C}(A)}{\mathbb{C}(B_i)} \tag{7}$$

where $\mathbb{C}(A)$ and $\mathbb{C}(B_i)$ stand for the raw count of game step $A$ and $B_i$ in the collected trajectories. The key intuition here being that state, action pairs that appear in a certain ordering in multiple trajectories are more likely to dependant on each other. Higher $\mathbb{P}(A \mid B_i)$ indicates the necessity of $A$ to $B$. The set of game steps with the highest $\mathbb{P}(A \mid B_i)$ is used to explain taking the action associated with game state $A$. For example, "take egg" ($A$) is required to "open egg" ($B$), and $\mathbb{P}(A \mid B) = 1$, hence "open egg" is used as a reason why action "take egg" must be taken first. The initial set of game states $\mathbf{X}$ is filtered into $\mathbf{X}_1$ by working backwards from the final goal state by finding the set of states that form the most likely chain of causal dependencies that lead to it. As shown in Figure 3, we obtain the explanation candidate game steps $\mathbf{X}_1$ by filtering all the possible game steps following current game step $A$ in the game logs with $\mathbb{P}(A \mid B_i) > p$, where $p$ is the threshold.

## A.7 Raw scores across Jericho supported games

Table 3: Raw scores across Jericho supported games. *Eps.* indicates scores averaged across the final 100 episodes and *Max* indicates the maximum score seen by the agent over the same period. We present results on *game and IM* reward.

| Exp. | TDQN | DRRN | HEX-RL | | Max |
|---|---|---|---|---|---|
| | | | *Game and IM* | | |
| Metric | Eps. | Eps. | Eps. | Max | - |
| zork1 | 9.9 | 24.6 | 30.2 | 40 | 350 |
| library | 6.3 | 17 | 13.8 | 21 | 30 |
| detective | 169 | 197.8 | 276.93 | 330 | 360 |
| balances | 4.8 | 10 | 10 | 10 | 51 |
| pentari | 17.4 | 27.2 | 44.7 | 60 | 70 |
| ztuu | 4.9 | **21.6** | 5.08 | 9 | 100 |
| ludicorp | 6 | 13.8 | 17.6 | 18 | 150 |
| deephome | 1 | 1 | 1 | 1 | 300 |
| temple | 7.9 | 7.4 | 7.58 | 8 | 35 |
| % compl. | 15.2 | 25.5 | **28.2** | **35.8** | 100 |

### A.8 Reward types

To alleviate the issue that rewards are sparse and often delayed, Ammanabrolu et al. defined an *intrinsic motivation* for the agent that leverages the knowledge graph being built during exploration. The motivation is for the agent to learn more information regarding the world and expand the size of its knowledge graph. They formally define $game\_and\_IM$ reward in terms of new information learned.

$$r_{\text{IM}_t} = \Delta(KG_{\text{global}} - KG_t) \tag{8}$$

where $KG_{\text{global}} = \bigcup\limits_{i=1}^{t-1} KG_i$ Here $KG_{\text{global}}$ is the set of all edges that the agent has ever had in its knowledge graph and the subtraction operator is a set difference.

### A.9 HEX-RL Architecture Hyperparameters

The additional hyperparamters used for training HEX-RL are detailed below, same with (Ammanabrolu et al., 2020). *graph dropout* and *mask dropout* are used for encouraging graph network to actually learn a sparse representation.

| Parameters | Value |
|---|---|
| buffer size | 40 |
| batch size | 16 |
| graph dropout | 0.2 |
| mask dropout | 0.1 |
| embedding size | 50 |
| hidden size | 100 |
| GAT embedding size | 25 |

### A.10 Task Performance

We plot the training reward curve for 9 games in Figure 7 and Figure 8. Reward curves are shown until they reach asymptotic performance, i.e. the number of steps until the score no longer increases. All agents were trained with the same number of steps (100,000) and the results at the end of this training is what is reported in Table 2.

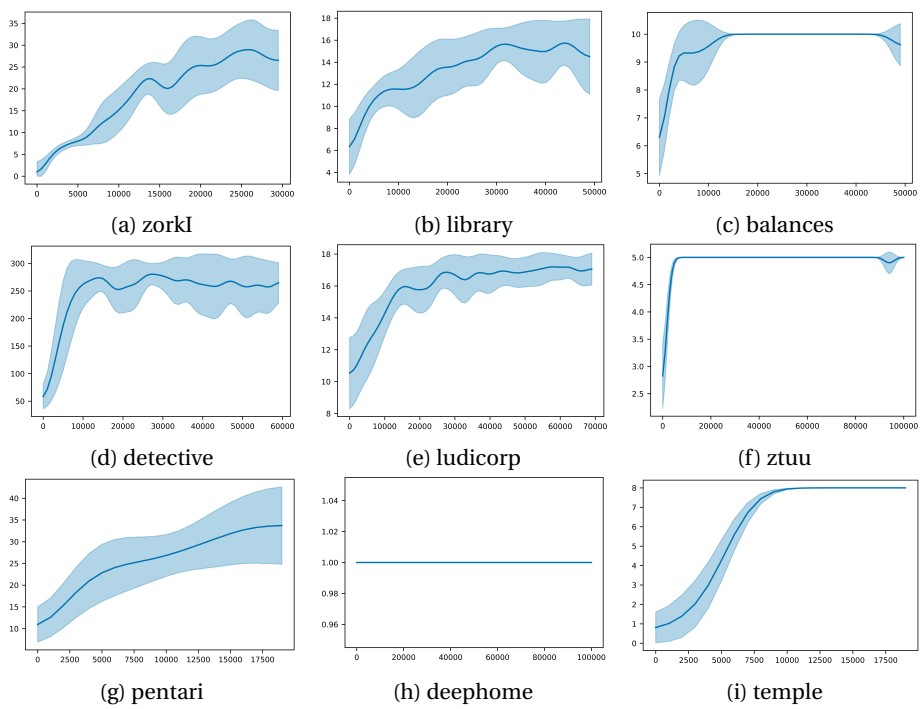

Figure 7: Eps. initial reward curves for the exploration strategies—*Game only* Reward

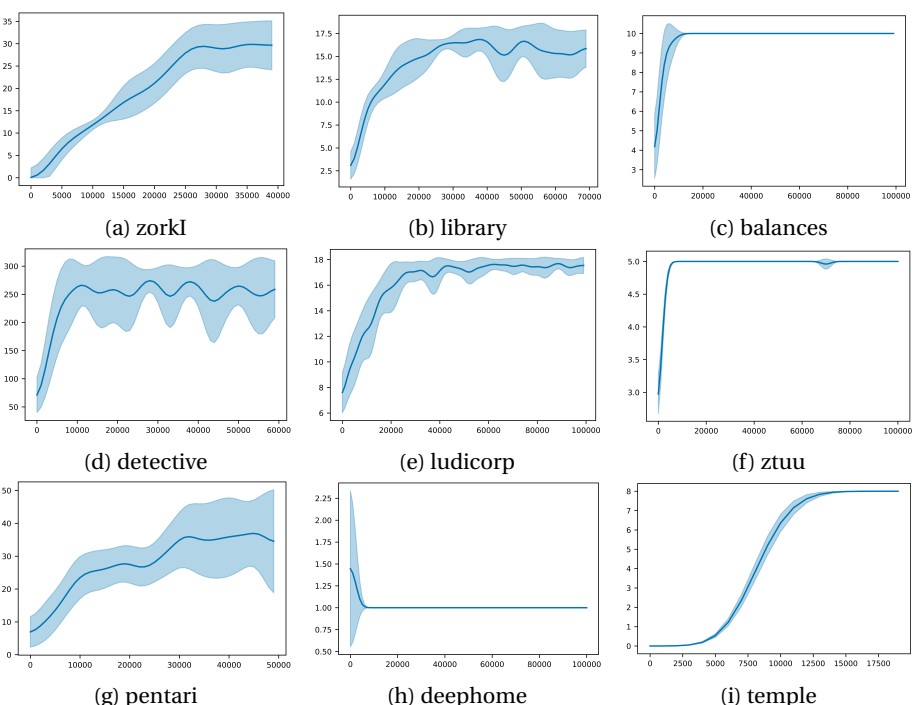

Figure 8: Eps. initial reward curves for the exploration strategies—*Game and IM* Reward

## A.11  Immediate Explanation Evaluation

We plot the immediate explanation evaluation result per game in Figure 9.

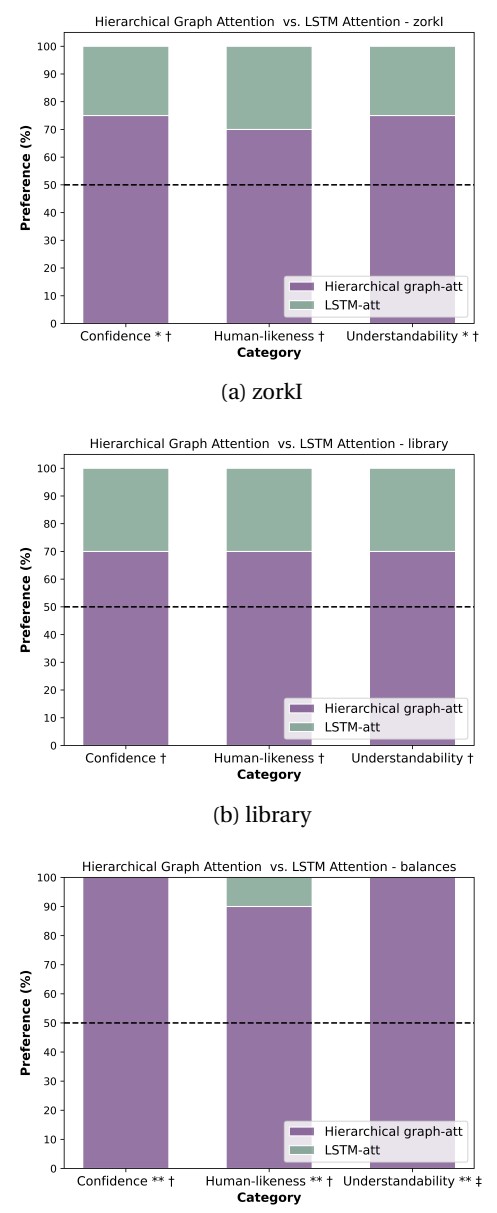

(a) zorkI

(b) library

(c) balances

Figure 9: Human evaluation results comparing Hierarchical Graph Attention vs. LSTM Attention, $*$ indicates $p < 0.05$, $**$ indicates $p < 0.01$, † indicates $\kappa > 0.2$ or fair agreement. ‡ indicates $\kappa > 0.4$ or moderate agreement.

## B Human Evaluation Details

### B.1 Immediate Explanation Evaluation

We firstly ask participants to read an interactive game description and then ask them to answer a set of questions about this game to make sure they are qualified. They will also play a demo of an interactive text game and answer a question based on the game they played. The details can be found in Figure 10 and Figure 11. These questions are designed to improve the quality of human evaluation. At least 5 participants give their preference for each explanation pair.

Each participant reads a randomly selected subset of 10 explanation pairs (drawn randomly from a pool totaling 60 explanation pairs), generated by Hierarchical Graph Attention and LSTM attention explanation on three games in the Jericho benchmark, *zork1*, *library*, and *balances*. The following three questions are asked,

- Which explanation makes you more confident that the agent made the right choice?
- Which explanation expresses more human-like thinking on the action choice?
- Which explanation makes you understand why the agent made the choice?

You will be asked to read a game trajectory of an **interactive fiction** game and then evaluate the explanations of choices in the game.

Please read this **interactive narrative game** description carefully!

*Balances* is the game in this study.

Description:
Balances is a relatively short, old-fashioned puzzle game set in the world of the Enchanter series and riffing on Spellbreaker in particular: the player must find scrolls, learn their spells, and cast them in order to collect white cubes. It shares a number of design characteristics with those games: rooms represent fairly large open spaces, there are more animal NPCs than humans to interact with, and a loose, playful approach to world-building means that the various areas don't have a great deal to do with one another.

>> We trained an AI agent to play the game and also automatically generate explanations for its choices. Your job is to read these stories and evaluate the **automatic explanations** by answering 3 questions.

You will first read one game step: a description and an action.
*Description* is to describe the room or location which the AI agent was in, and *action* is the agent's choice based on the descriptions.
Then you will **read two explanations** of the above action. 3 questions will be asked for PAIRWISE COMPARISON:
1. Which explanation makes you more **confident** that the agent made the right choice?
2. Which explanation expresses more **human-like** thinking on the action choice?
3. Which explanation makes you **understand** why the agent made the choice?

----------------------------------
*What will you do in* *Balances*?
**(Multiple choices, select all that apply)**

☐ Get treasure

☐ Find scrolls

☐ Find murderers

☐ Learn spells

Figure 10: Screenshot of the human study instruction—game description.

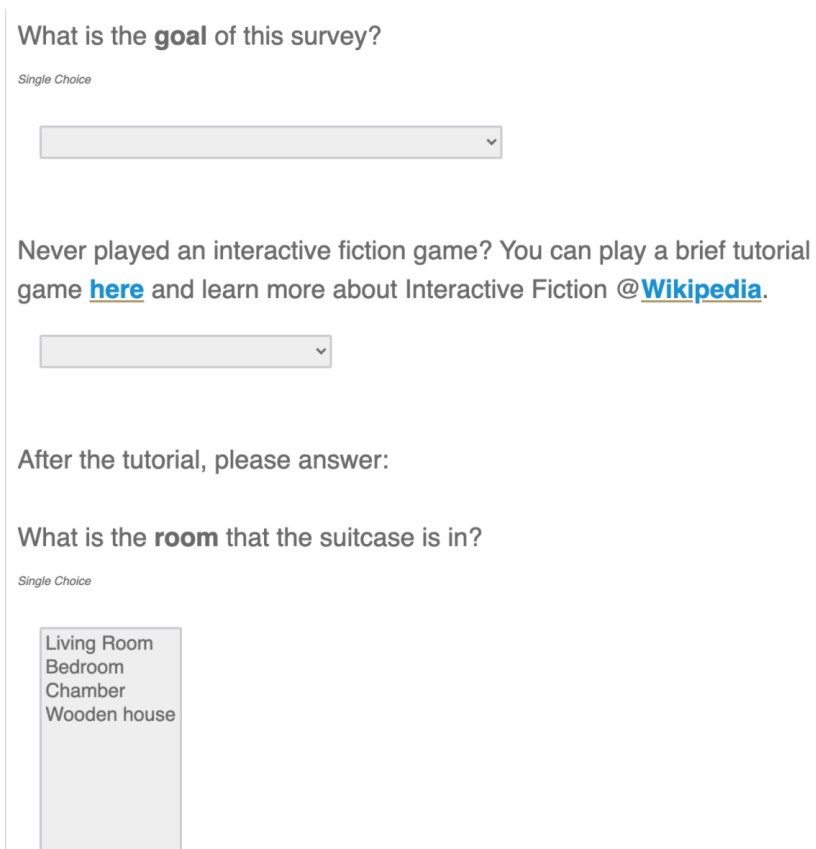

Figure 11: Screenshot of the human study instruction—task description.

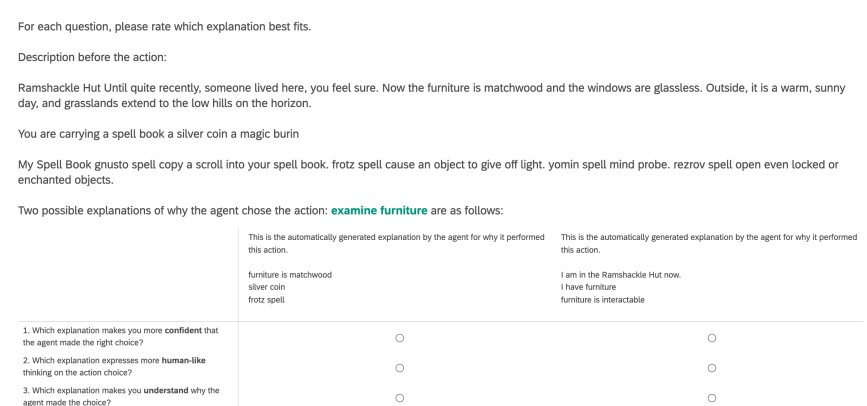

Figure 12: Screenshot of the human study instruction.

## B.2 Immediate vs. Temporal Explanation Evaluation

Participants first read the full trajectory of the game (Figure 13) combined with step-by-step immediate explanations, along with summary of the game goal, and indicate how much they agree with the five statements on a Likert scale (Figure 14). The following five statements are used in human study.

- I am confident that I can get the same score as the agent when following this explanation.
- This explanation look like it was made by human.
- This explanation is easy to understand.
- I am able to understand why the agent takes this particular sequence of actions given what I know about the goal.
- This explanation is easy to read.

At least 5 crowd workers rated each explanation.

Please read this transcript of a player playing *Library*:

**Goal**: Take the book

**Description**:
Second Floor Stacks
This cavernous room is filled with shelves as far as the eye can see. A doorway to the east is labelled "Computer Room", and the stairwell lies to the north. The door is unlocked but shut.

*Action: undo door*

--------------------
**Description**:
You open the rare books door.

*Action: south*

**Description after taking the above action**:
Rare Books Room The shelves are nearly bare, although there is a complete set of the "New ork Times", a box labeled "Avalon", and several biographies of various computer game authors. The door out is to the north You can see a biography of Graham Nelson here.

--------------------
**Description**:
Rare Books Room
The shelves are nearly bare, although there is a complete set of the "New ork Times", a box labeled "Avalon", and several biographies of various computer game authors. The door out is to the north You can see a biography of Graham Nelson here.

*Action: take all*

**Description after taking the above action**:
biography of Graham Nelson Taken. Your score has just gone up by five points.

--------------------
Can you **summarize** what is happening in this transcript in less than 100 words?

Figure 13: Screenshot of Immediate vs. Causal Explanation Evaluation —Text Summary.

Please answer the following questions about the game transcript given what you know about the goals of the game.

|  | Strongly Agree | Agree | Undecided | Disagree | Strongly Disagree |
|---|---|---|---|---|---|
| 1. I am **confident** that I can get the same score as the agent when following this explanation. | ○ | ○ | ○ | ○ | ○ |
| 2. This explanation look like it was made by **human**. | ○ | ○ | ○ | ○ | ○ |
| 3. This explanation is easy to **understand**. | ○ | ○ | ○ | ○ | ○ |
| 4. I am able to understand why the agent takes this particular sequence of actions given what I know about the goal. | ○ | ○ | ○ | ○ | ○ |
| 5. This explanation is easy to **read**. | ○ | ○ | ○ | ○ | ○ |

Figure 14: Screenshot of Immediate vs. Causal Explanation Evaluation—Likert Scale.

We also plot the causal explanation ablation study result per game in Figure 15 and Figure 16.

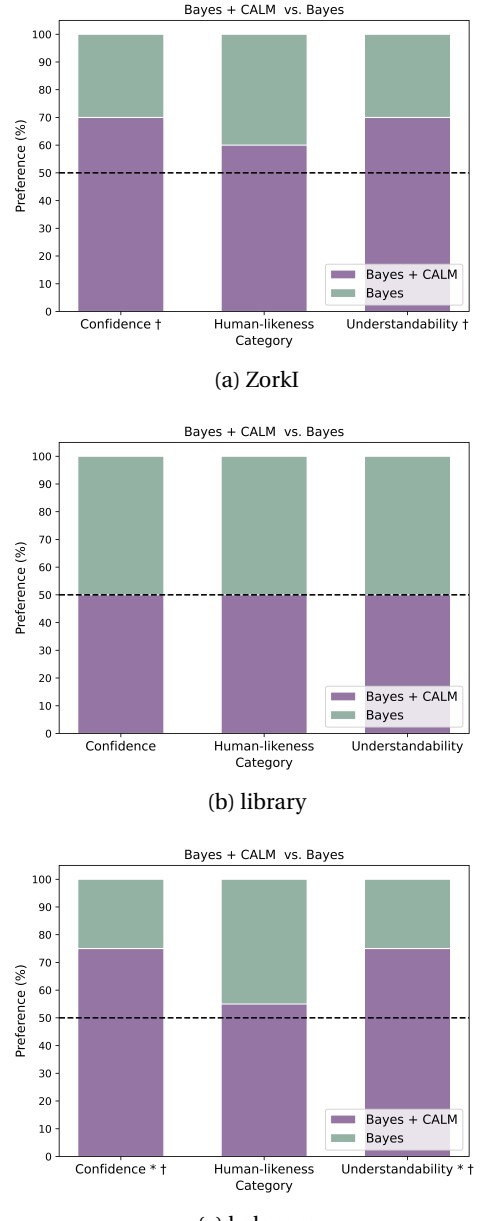

(a) ZorkI

(b) library

(c) balances

Figure 15: Human evaluation results on ablation study, $*$ indicates $p < 0.05$, $\dagger$ indicates $\kappa > 0.2$ or fair agreement.

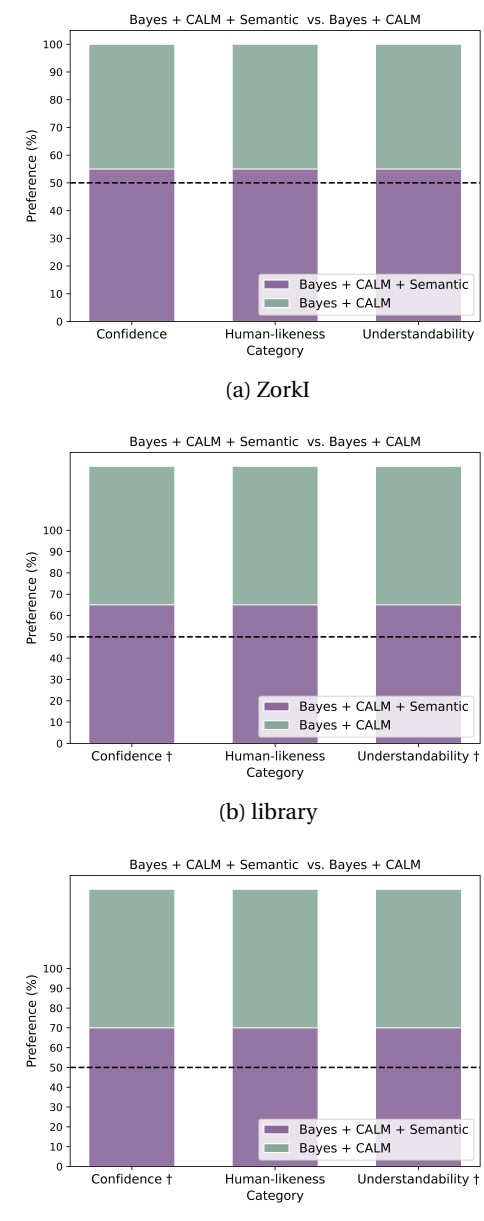

(a) ZorkI

(b) library

(c) balances

Figure 16: Human evaluation results on ablation study, ∗ indicates $p < 0.05$, † indicates $\kappa > 0.2$ or fair agreement.

## C  Explanations Analysis

In this section we provide descriptions of each of the games as well as qualitative samples of trajectories in the format shown to the human participants. We pick out two types of examples where the HEX-RL explanations of the trajectories were rated highly by the human participants and rated poorly in terms of the quality of the explanations—attempting to analyze the failure cases of our method.

### C.1  Text Game Descriptions

We list the descriptions[5] of our selected text games for a better undestanding of our explanations and game trajectories in Appendix C.2 and C.3.

1. *zork1*: Many strange tales have been told of the fabulous treasure, exotic creatures, and diabolical puzzles in the Great Underground Empire. As an aspiring adventurer, you will undoubtedly want to locate these treasures and deposit them in your trophy case. Zork creates a wondrous, magical realm that is a veritable feast for the imagination. You find that you have stumbled upon the ancient ruins of a vast empire lying far underground. Yes, you will find many more treasures for your trophy case. But to do so, you will have to search far and wide, solve diabolical puzzles, and defend your treasures (and yourself!) from a few very nasty characters... and one monster, a vicious GRUE that lurks in the dark!

2. *Library*: Escape the library using knowledge and objects found in and about the library.

3. *Balances*: Balances is a relatively short, old-fashioned puzzle game set in the world of the Enchanter series and riffing on Spellbreaker in particular: the player must find scrolls, learn their spells, and cast them in order to collect white cubes. It shares a number of design characteristics with those games: rooms represent fairly large open spaces, there are more animal NPCs than humans to interact with, and a loose, playful approach to world-building means that the various areas don't have a great deal to do with one another.

### C.2  Immediate Explanations Analysis

Table 4 shows some example trajectories with its immediate explanations where Hierarchical Graph Attention Explanations did a better job than LSTM Attention. We can see when the player interacts with objects or the subset of sub knowledge graph can explain the action, Hierarchical Graph Attention is able to present a high-quality immediate explanation.

Table 5 shows some example trajectories with its immediate explanations where HEX-RL fails to produce correct immediate explanations. We see that when there is no clear clue in the observations, i.e. the potential gold truth explanation is does not belong knowledge graph, HEX-RL fails to produce immediate explanations.

Hence, we conclude that the performance of immediate explanations is limited: (1) to the cases in which explanation is directly linked to the one of the facts we choose to extract from the knowledge graph, such as object/location information; and (2) by the relative error of knowledge graph extraction models themselves.

### C.3  Temporally Extended Explanations Analysis

Table 6, 7 and 8 show example game trajectories with its corresponding temporally extended explanations for game *zork1*, *library* and *balances*. HEX-RL finds the subset of states that are most crucial to achieving the task goals.

We also show one trajectory where HEX-RL fails to produce a high-quality temporally extended explanations in *zork1*. Here, in order to achieve the goal "open window", the player has to get to the location "North of House". However, there exists too many different combinations of actions,

---

[5]https://ifdb.org/

i.e. branching paths, which the player can take to go to "North of House". Bayesian State Filter thus fails to find the most essential steps for achieving this goal.

The performance of the temporally extended explanation system is hindered by the complexity of the game—the more possible ways of achieving the same overall goal, the harder it is to generate an explanation for why the agent has chosen to perform an action to achieve the goal.

Table 4: Example game trajectories with its immediate explanations – Part 1.

[*zork1*]Description before the action:
North of House
You are facing the north side of a white house.
There is no door here, and all the windows are boarded up.
To the north a narrow path winds through the trees.
**Action: go north**

| LSTM:
You are facing the north side of a white house. | Hierarchical Graph :
Trees is north of north of house. |
| --- | --- |

[*zork1*]Description before the action:
Kitchen
You are in the kitchen of the white house.
A table seems to have been used recently for the preparation of food.
A passage leads to the west and a dark staircase can be seen leading upward.
A dark chimney leads down and to the east is a small window which is open.
On the table is an elongated brown sack, smelling of hot peppers.
A bottle is sitting on the table. The glass bottle contains A quantity of water.
**Action: open bottle**

| LSTM:
On the table is an elongated brown sack,
smelling of hot peppers. | Hierarchical Graph :
I am in the Kitchen now.
I have bottle.
Bottle is interactable. |
| --- | --- |

[*zork1*]Description before the action:
Up a Tree
You are about 10 feet above the ground nestled among some large branches.
The nearest branch above you is above your reach.
Beside you on the branch is a small birds nest.
In the birds nest is a large egg encrusted with precious jewels,
apparently scavenged by a childless songbird.
The egg is covered with fine gold inlay, and ornamented in lapis lazuli and motherofpearl.
Unlike most eggs, this one is hinged and closed with a delicate looking clasp.
The egg appears extremely fragile.
**Action: take egg**

| LSTM:
In the birds nest is a large egg encrusted
with precious jewels. | Hierarchical Graph :
egg is interactable. |
| --- | --- |

[*library*]Description before the action:
Rare Books Room
The shelves are nearly bare, although there is a complete set of the "New ork Times",
a box labeled "Avalon", and several biographies of various computer game authors.
The door out is to the north You can see a copy of "Debt of Honor" here.
**Action: use key**

| LSTM:
The shelves are nearly bare. | Hierarchical Graph :
I have key.
key is interactable. |
| --- | --- |

Table 5: Example game trajectories with its immediate explanations – Part 2.

| [*zork1*]Description before the action: |
|---|
| Forest Path |
| This is a path winding through a dimly lit forest. |
| The path heads northsouth here. |
| One particularly large tree with some low |
| branches stands at the edge of the path. |
| **Action: go up** |

| LSTM: | Hierarchical Graph : |
|---|---|
| This is a path winding through a dimly lit forest. | tree is interactable. |

| [*library*]Description before the action: |
|---|
| Rare Books Room |
| The shelves are nearly bare, although there is a complete set of the "New ork Times", |
| a box labeled "Avalon", and several biographies of various computer game authors. |
| The door out is to the north |
| You can see a biography of Graham Nelson here. |
| **Action: go west** |

| LSTM: | Hierarchical Graph : |
|---|---|
| The door out is to the north. | I am in the Rare Books now. |

| [*balances*]Description before the action: |
|---|
| Grasslands, near Hut |
| The grasslands sway over low hills in all directions it is a peaceful wilderness, |
| broken only by this hut and a faint path to the north. |
| You are carrying a cedarwood box, a spell book, a silver coin and a magic burin. |
| **Action: go north** |

| LSTM: | Hierarchical Graph : |
|---|---|
| You are carrying a cedarwood box. | I am in the near Hut now. |

Table 6: Example game trajectories for *zork1* with its temporally extended explanations – part 1.

| **Goal: Take Egg** |
| --- |
| Description:
Kitchen
You are in the kitchen of the white house. A table seems to have been used recently for the preparation of food. A passage leads to the west and a dark staircase can be seen leading upward. A dark chimney leads down and to the east is a small window which is open. On the table is an elongated brown sack, smelling of hot peppers. Sitting on the kitchen table is A glass bottle The glass bottle contains A quantity of water
Action: east |
| Description:
Behind House
You are behind the white house. A path leads into the forest to the east. In one corner of the house there is a small window which is open.
Action: east |
| Description:
North of House
You are facing the north side of a white house. There is no door here, and all the windows are boarded up. To the north a narrow path winds through the trees.
Action: north |
| Description:
Forest Path
This is a path winding through a dimly lit forest. The path heads northsouth here. One particularly large tree with some low branches stands at the edge of the path.
Action: up |
| Description:
Up a Tree
You are about 10 feet above the ground nestled among some large branches. The nearest branch above you is above your reach. Beside you on the branch is a small birds nest. In the birds nest is a large egg encrusted with precious jewels, apparently scavenged by a childless songbird. The egg is covered with fine gold inlay, and ornamented in lapis lazuli and mother of pearl. Unlike most eggs, this one is hinged and closed with a delicate looking clasp. The egg appears extremely fragile.
Action: take egg |
| Description:
Taken. |

| **Temporally Extended Explanations** |
| --- |
| Description:
Forest Path
This is a path winding through a dimly lit forest. The path heads northsouth here. One particularly large tree with some low branches stands at the edge of the path.
Action: up |
| Description:
Up a Tree
You are about 10 feet above the ground nestled among some large branches. The nearest branch above you is above your reach. Beside you on the branch is a small birds nest. In the birds nest is a large egg encrusted with precious jewels, apparently scavenged by a childless songbird. The egg is covered with fine gold inlay, and ornamented in lapis lazuli and mother of pearl. Unlike most eggs, this one is hinged and closed with a delicate looking clasp. The egg appears extremely fragile.
Action: take egg |
| Description after taking the above action:
Taken. |

Table 7: Example game trajectories for *library* with its temporally extended explanations.

| **Goal: Take the book** |
| --- |
| Description:
Second Floor Stacks
This cavernous room is filled with shelves as far as the eye can see. A doorway to the east is labelled "Computer Room", and the stairwell lies to the north. A heavy wooden door bars the way to the rare books room. A sign indicates that the key may be obtained from the circulation desk.
Action: undo door with key |
| Description:
You unlock the rare books door.
Action: east |
| Description:
Computer Lab
This entire room hums with the combined output of several dozen computers. All of them are hopelessly out of date, but at least theyre still good for word processing. A bank of printers on one side of the room frantically churns out paper. You can see a technician here.
Action: west |
| Description:
Second Floor Stacks
This cavernous room is filled with shelves as far as the eye can see. A doorway to the east is labelled "Computer Room", and the stairwell lies to the north. The door is unlocked but shut.
Action: undo door |
| Description:
You open the rare books door.
Action: south |
| Description:
Rare Books Room
The shelves are nearly bare, although there is a complete set of the "New ork Times", a box labeled "Avalon", and several biographies of various computer game authors. The door out is to the north You can see a biography of Graham Nelson here.
Action: take all |

| **Temporally Extended Explanations** |
| --- |
| Description: Second Floor Stacks
This cavernous room is filled with shelves as far as the eye can see. A doorway to the east is labelled "Computer Room", and the stairwell lies to the north. The door is unlocked but shut.
Action: undo door |
| Description: You open the rare books door.
Action: south
Description after taking the above action: Rare Books Room The shelves are nearly bare, although there is a complete set of the "New ork Times", a box labeled "Avalon", and several biographies of various computer game authors. The door out is to the north You can see a biography of Graham Nelson here.
... |
| Description: Rare Books Room
The shelves are nearly bare, although there is a complete set of the "New ork Times", a box labeled "Avalon", and several biographies of various computer game authors. The door out is to the north You can see a biography of Graham Nelson here.
Action: take all
Description after taking the above action: biography of Graham Nelson Taken.
Your score has just gone up by five points. |

Table 8: Example game trajectories for *balances* with its temporally extended explanations.

**Goal: Examine Pile**

Description: Grasslands, near Hut
The grasslands sway over low hills in all directions it is a peaceful wilderness, broken only by this hut and a faint path to the north. You can see a spell book here.
Action: east

Description: Ramshackle Hut
Until quite recently, someone lived here, you feel sure. Now the furniture is matchwood and the windows are glassless. Outside, it is a warm, sunny day, and grasslands extend to the low hills on the horizon.
Action: take inventory

Description: The wooden furniture is already on the floor.
Action: west

Description: Grasslands, near Hut
The grasslands sway over low hills in all directions it is a peaceful wilderness, broken only by this hut and a faint path to the north. You can see a spell book here.
Action: east

Description: Ramshackle Hut
Until quite recently, someone lived here, you feel sure. Now the furniture is matchwood and the windows are glassless. Outside, it is a warm, sunny day, and grasslands extend to the low hills on the horizon.
Action: west

Description: Grasslands, near Hut
The grasslands sway over low hills in all directions it is a peaceful wilderness, broken only by this hut and a faint path to the north. You can see a spell book here.
Action: east

Description: Ramshackle Hut
Until quite recently, someone lived here, you feel sure. Now the furniture is matchwood and the windows are glassless. Outside, it is a warm, sunny day, and grasslands extend to the low hills on the horizon.
Action: examine book

Description: My Spell Book gnusto spell copy a scroll into your spell book. frotz spell cause an object to give off light. yomin spell mind probe. rezrov spell open even locked or enchanted objects.
Action: examine furniture

Description: You see nothing special about the wooden furniture.
Action: west

Description: Grasslands, near Hut
The grasslands sway over low hills in all directions it is a peaceful wilderness, broken only by this hut and a faint path to the north.
Action: north

Description: Pocket Valley
A pleasant pocket valley in the grassy hills, through which a trail runs northtosouth. There is a chestnut horse here, munching on a pile of oats.
Action: examine pile

**Temporally Extended Explanations**

Description: Ramshackle Hut
Until quite recently, someone lived here, you feel sure. Now the furniture is matchwood and the windows are glassless. Outside, it is a warm, sunny day, and grasslands extend to the low hills on the horizon.
Action: examine book

Description: My Spell Book gnusto spell copy a scroll into your spell book. frotz spell cause an object to give off light. yomin spell mind probe. rezrov spell open even locked or enchanted objects.
Action: examine furniture
Description after the action:
Searching through the furniture, which is good for nothing but firewood now, you come across an old cedarwood box, which you pick up for a closer look. Your score has just gone up by five points.
...

Description: Pocket Valley A pleasant pocket valley in the grassy hills, through which a trail runs north-tosouth. There is a chestnut horse here, munching on a pile of oats.
Action: examine pile

Table 9: Example game trajectories for *zork1* with its temporally extended explanations – part 2.

| **Goal: Open Window** |
|---|
| Description:
Forest Path
This is a path winding through a dimly lit forest.
The path heads northsouth here.
One particularly large tree with some low branches stands at the edge of the path.
Action: go east |
| Description:
Forest This is a dimly lit forest, with large trees all around.
Action: go south |
| Description:
Clearing
You are in a small clearing in a well marked forest path that extends to the east and west.
Action: go west |
| Description:
Behind House
You are behind the white house. A path leads into the forest to the east. In one corner of the house there is a small window which is open.
Action: go west |
| Description:
Clearing
You are in a small clearing in a well marked forest path that extends to the east and west.
Action: go north |
| Description:
West of House
You are standing in an open field west of a white house, with a boarded front door. There is a small mailbox here.
Action: go north |
| Description:
North of House
You are facing the north side of a white house. There is no door here, and all the windows are boarded up. To the north a narrow path winds through the trees.
Action: go east |
| Description:
Behind House
You are behind the white house. A path leads into the forest to the east. In one corner of the house there is a small window which is slightly ajar.
Action: open window |

| **Temporally Extended Explanations** |
|---|
| Description:
Behind House
You are behind the white house. A path leads into the forest to the east. In one corner of the house there is a small window which is slightly ajar.
Action: open window |

## D   Broader Impacts

The ability to explain the reinforcement learning actions in text games has downstream applications beyond understanding how to play text games. Text games are regarded as simplified analogues for systems capable of long-term dialogue with humans, such as in assistance with planning complex tasks, and also discrete planning domains such as logistics. Our work is applicable to provide understanding of domains where change in the world is affected via language. Systems prone to error, both in performing a given task as well as in explaining them, should not be relied upon in more critical applications.