# OpenReview forum: "Inherently Explainable Reinforcement Learning in Natural Language"
_NeurIPS.cc/2022/Conference — NeurIPS 2022 Accept_

### Official Review · Reviewer_H13U · 2022-06-19

**Rating:** 4
**Confidence:** 3
**Soundness:** 3 good
**Presentation:** 3 good
**Contribution:** 2 fair

**Summary:**

This paper focuses on learning to play text adventure games using RL while producing immediate step by step explanation, as well as a trajectory level explanation using a KG based state representation combined with Hierarchical Graph Attention, which they dubbed as HEX-RL. Experiments show comparable performance for playing text-based games compared to SOTA while providing improvement in terms of explainability compared to baseline.

**Questions:**

1. What is the full action space? Some discussion about it in main paper would be helpful, Also how this method can be applied in settings with large action space.

**Strengths And Weaknesses:**

Pros:

* Paper is very well written and motivated.
* Authors provide good review of text games' literature.
* Human evaluation and the metic for comparing intermediate vs trajectory comparison (i.e., Goal context) seem solid and interesting.

Cons:

* The novelty is a bit low. The whole ideas used here for immediate explanation and trajectory level explanation are not totally new and novelty is limited.
In terms of immediate explanation, the idea of using Hierarchical Graph Attention for explainability is not new and a similar approach has been proposed in [Xu et al.]. Same applies In terms of providing trajectory level explanation. Even though there are some novelty in combining three steps filtering approach, but each of them separately is not novel.  Also authors provide an ablation study  in Appendix C which was useful experiment, but I am not fully convinced if those three filters are all necessary. I think it would have been interesting to see the performance of just a larger language model (i.e., a larger CALM) for filtering or a fine tuned version to further simplify the design of filtering.
* I am not fully convinced of this claim in the introduction-- "study against strong baselines that shows that our agent generates significantly improved explanations"; would have been nice to see other closely related baseline for Immediate explanation Evaluation such as [Xu et al.]

Admittedly, I see novelty in combining two papers [Xu et al. + Ammanabrolu et al.] but I think overall novelty is limitted.

---

> ### Author Response · Authors · 2022-08-02
> **Clarifying novelty and baselines**
>
> We would first like to thank the reviewer for their thoughtful comments and time. We will clarify claims below.
>
> 1. _''The novelty is a bit low. The whole ideas used here for immediate explanation and trajectory level explanation are not totally new and novelty is limited. In terms of immediate explanation, the idea of using Hierarchical Graph Attention for explainability is not new and a similar approach has been proposed in [Xu et al.]. The same applies In terms of providing a trajectory-level explanation. ''_
>
> - To clarify the novelty of our approach - we are the first to use knowledge graph attention-based attribution to explain actions in such grounded environments.
>
>     Even though Qbert (Ammanabrolu et al.)  and SHA-KG (Xu et al.) are both knowledge graph-based agents, these architectures do now allow for as fine-grained attention-based attribution as our architecture does---e.g. Q*BERT does not use relationship information in their policy and SHA-KG averages attention across large portions of the graph and is unable to point to specific triples in its KG representation to explain an action.
>
> 2. _''Authors provide an ablation study in Appendix C which was useful experiment, but I am not fully convinced if those three filters are all necessary. I think it would have been interesting to see the performance of just a larger language model (i.e., a larger CALM) for filtering or a fine tuned version to further simplify the design of filtering.''_
>
> - In terms of temporal explanations, the reason why we did not use a *single* general model to produce temporally-extended explanations in this paper is that different filters serve different functions.
>
>     - Bayesian filter helps to form the most likely chain of causal dependencies that lead to a state. It only considers causal dependencies in this specific game without any semantic relationship between steps.
>     - CALM filters down the set of important states by finding the states that have corresponding actions that a human player would be more likely to perform. Since CALM is trained on human transcripts, it considers causal dependencies from the human perspective.
>     - The semantic State-Action Filter takes semantic relationships into consideration.
>
>     These three filters serve different requirements of temporally-extended explanations. One can use any combination of them for serving different functions. For example, we can produce temporally-extended explanations about causal with the Baysian filter only.
>
>     To the best of our knowledge, each of these filters themselves has never been used for the purposes of reducing an agent's trajectory to provide more useful temporal explanations. We contend that all steps are individually novel as well as their combination.
>
> 3. _''I am not fully convinced of this claim in the introduction-- "study against strong baselines that shows that our agent generates significantly improved explanations"; would have been nice to see other closely related baseline for Immediate explanation Evaluation such as [Xu et al.]''_
>
> - Our paper is the first to use knowledge graph attention-based attribution to explain actions in such grounded environments, hence there immediate prior works to compare to. We have adapted prior works and devised strong baselines here to compare to.
>
>     As stated before, SHA-KG  [Xu et al.] has no component to provide human-readable explanations. They average attention across large portions of the graph and so are unable to point to specific triples in its KG representation to explain an action.  Simply put, their agent can show whether a relatively large subgraph was relevant for the current action but cannot provide a readable explanation for it. They consequently do not validate or test the utility of their explanations via participant study either.
>
> 4. _''What is the full action space? Some discussion about it in main paper would be helpful, Also how this method can be applied in settings with large action space.''_
>
> - Thanks for pointing this out.
>
>     This is an established action space used by the literature for the Jericho benchmark:
>
>     ''Jericho provides the capability to extract game-specific vocabulary and action templates. These templates contain up to two blanks, so a typical game with 200 templates and a 700 word vocabulary yields an action space of O(TV^2 ) ≈ 98 million, three orders of magnitude smaller than the 240-billion space of 4-word actions using vocabulary alone.''
>
>     We have relegated this discussion to Appendix A.1  in the interest of space.

---

> ### Author Response · Authors · 2022-08-08
> **Gentle reminder that the discussion period closes soon**
>
> This is a gentle reminder that the discussion period closes soon. We have attempted to address all concerns raised, summarized the rationale for doing so and modified the paper itself. We would be greatly encouraged if the reviewers either raised their scores or engaged in further discussion with us. Thanks a lot.

---

### Official Review · Reviewer_SZnA · 2022-06-30

**Rating:** 7
**Confidence:** 4
**Soundness:** 2 fair
**Presentation:** 4 excellent
**Contribution:** 3 good

**Summary:**

This paper explores how to leverage graph knowledge within an RL agent to provide an a-posteriori explanation of the agent's actions.
The authors first motivate their approach through a well-written introduction clearly stating the research objective.
They then detail the underlying neural mechanisms and the different filtering layers to post-process the graph and generate the language explanation.
Finally, the authors provide two qualitative analyses: a performance benchmark on the agent abilities and a human survey on the quality of the explanation.

**Questions:**

 I have four main concerns:
 A) the lack of model ablation
 B) the lack of discussion about the method limitation (which may be linked to ablation)
 C) the lack of qualitative results to give a better intuition of the method
 D) the lack of information for reproducibility

A) The model is based on multiple design choices, yet none of this choice is really discussed. First, both the agent performance and explainability skills are based on the structure of the graph. Therefore, changing the graph, e.g., hiding/fusing subgraphs, or injecting noise, would be of interest to see how the model depends on the expert choice. It exists a discussion in Appendix C (the impact of the filtering), which is never mentioned in the paper, I would recommend discussing it in the core paper. The impact of some hyperparameters (e.g., l193, trajectories) are also not discussed while they may be of paramount importance. Finally, in 4.2, intermediate baseline models, LSTM + slot-filling without KG would allow to see whether the model is really more understandable with a KG. Here, it is a bit unfair to compare an unconstraint LSTM model with a grammar-based model while assessing for clarity.

B) It may be the most significant paper weakness: positivism. In other words, the model limitations are never discussed, and the analysis remains purely quantitive without any hints for future direction. How hard is the training/tuning? What are the model mistakes? Would it be possible to compute some language statistics? And importantly, how satisfactory are the explanations? So far, the authors only provide an absolute goal score. However, I would tend to think that the current definition may be too light to apprehend the model performance correctly. Something like: would you be able to reconstruct the sequence of actions given the explanation? Are you satisfied with the current explanation? Furthermore, I would also split the analysis into successful/unsuccessful trajectories to better disentangle the soundness and correctness of the explanation. A perfect example would be: the agent was wrong, yet, did its choices make sense? In any case, happy to further discuss this point!

C) So far, the reader can't have any intuition on the model skills. The only actual example is in the user interface in the appendix. Please provide some [5+] cherry-picked text-game trajectories (positive and negative!). This is very useful when supporting l367, for instance.

D) Please add a table with all model parameters in the appendix and [optional] a training curve. I just cannot accept the paper without such things. For instance, the paper does not even provide the discount factor, which is not acceptable from an RL perspective. Please also add more content on the game description in the appendix, e.g. for each game, graph statistics, game interest description etc. Please add the std in Tab2; max is less interesting.

So again, the paper has many merits. I am leaning toward acceptance despite some potential improvement. However, I cannot accept it without having C and D solved and having a conversation on the model limitation.

**Strengths And Weaknesses:**

First of all, I want to felicitate the authors for the quality of the writing and their efforts in explaining the model and research direction. Independently of the paper's content, it was a pleasant and easy read.
Therefore, the research direction is well-stated. I also appreciated the absence of over-claiming in the paper... except for the title, which incorrectly mentions natural language; while it is templated language. Please remove the word 'natural' from the title.
The related work is a bit short, and I would recommend adding a small paragraph on the different research directions in text games that have been pursued, e.g., continual learning [1], action pruning [2], and many others.
Section 3 gives a good understanding of the method. Although it requires a bit of engineering, they are consistent, and the method remains a promising proof of concept. I would recommend adding some module names in Figure2 (middle) in addition to the network names; otherwise, it is hard to follow. I have a few concerns and questions about the model, which I will detail later.
In the experimental section, the authors correctly split the performance and the explainability. The human evaluation seems to have been correctly performed, which is not too common in the ML literature.
Overall, I got convinced by the approaches, and despite some task-specific design (especially in the filter choice and in the knowledge subgraph), the method seems extendable to other settings.

Overall, I am quite positive about the paper, and it is solid enough to be accepted. However, there are still some critical points I would like to discuss which restrain me from giving a high score.


[1] Shuster, Kurt, et al. "Deploying lifelong open-domain dialogue learning." arXiv preprint arXiv:2008.08076 (2020).
[2] Zahavy, Tom, et al. "Learn what not to learn: Action elimination with deep reinforcement learning." Advances in neural information processing systems 31 (2018).

---

> ### Author Response · Authors · 2022-08-02
> **Addressing some weaknesses and concern**
>
> We thank the reviewer for their time and effort and are encouraged by their analysis of this work's significance. We will attempt to address a couple of the weaknesses pointed out.
>
> 1. _''The related work is a bit short, and I would recommend adding different research directions in text games that have been pursued.''_
>
> - Thanks for your suggestion. We have revised our work and added as many additional citations as possible given space considerations.
>
> 2. _''I would recommend adding some module names in Figure 2 in addition to the network names.''_
>
> - Thanks for this suggestion. We added more module names in Figure 2 to make it easier to follow in revision.
>
> 3. _''It exists a discussion in Appendix C (the impact of the filtering), which is never mentioned in the paper.''_
>
> - Thanks for pointing this out.  We added the summary to **Line 256**.
>
> 4. _''The impact of some hyperparameters (e.g., l193, trajectories) are also not discussed while they may be of paramount importance.''_
>
> - Thanks for pointing this out. Our initial experiments suggested that the larger the number of trajectories we use (l193), the more accurate the Bayesian State Filter. We will revise our appendix to include a discussion regarding our hyperparameter choices. We further note that we report these parameters in **Appendix A.4**.
>
> 5. _''...unfair to compare an unconstraint LSTM model with a grammar-based model while assessing for clarity.''_
>
> - In order to have a fair comparison with the LSTM model, we extract the most important substring in the observations through LSTM attention and then use substrings of the human-written observation which contain those words to create an explanation akin to a slot filler. The LSTM model is therefore not unconstrained and we believe it aligns with the reviewer’s suggestion.
>
>     We will also note that many human participants state this LSTM explanation is more natural sounding than our immediate explanation from KG with a grammar-based model.
>
> 6. _''... the model limitations are never discussed, and the analysis remains purely quantitive without any hints for future direction.''_
>
> - Thanks for this great suggestion. As mentioned in the old paper, as our system relies on graph hierarchical graph attention to generate immediate explanations, we are limited to providing explanations on systems affected by natural language.
>
>     We added more limitations according to your advice to the *Evaluation* section (4.2 and 4.3) of the current revision.
>
>     As our system relies on graph hierarchical graph attention to generating immediate explanations, a well-trained knowledge graph representation module of the world knowledge is required. Hence, our model is limited to providing explanations on systems affected by natural language.
>
> 7. _''...how satisfactory are the explanations?...I would tend to think that the current definition may be too light to apprehend the model performance correctly. ''_
>
> - We asked human participants whether or not the agent's explanation was good enough to enable them to play the game with equivalent skill and asked the participant to condense this judgment down into a goal score. We believe this phrasing serves the same purpose as asking them to reconstruct the sequence of actions as the reviewer has suggested in addition to their satisfaction with the explanation.
>
> 8. _''... I would also split the analysis into successful/unsuccessful trajectories to better disentangle the soundness and correctness of the explanation.''_
>
> - Thanks for this advice. We added more examples and analysis in **Appendix D.2**.
>
>     We conclude that the performance of immediate explanations is limited:
>     - to the cases in which explanation is directly linked to the one of the facts we choose to extract from the knowledge graph, such as object/location information; and
>     - by the relative error of knowledge graph extraction models themselves.
>
> 9. _''Please provide text-game trajectories (positive and negative!).''_
>
> - Thanks for this advice. We added more examples and analysis in **Appendix D.2 and D.3**.
>
> 10. _''Please add a table with all model parameters in the appendix and a training curve. ''_
>
> - We added more model parameters in **Appendix A.2, A.3, and A.9**. The training curve was presented in Appendix A.10.
>
> 11. _''The paper does not even provide the discount factor, which is not acceptable from an RL perspective. ''_
>
> - We used the same hyper-parameters with Ammanabrolu et al.. We added more model parameters including the discount factor in **Appendix A.2**.
>
> 12. _''...add more content on the game description in the appendix.''_
>
> - We added game descriptions in Appendix D.1. Game statistics has been shown in Table 2.
>
> 13. _''Please add the std in Tab2''_
>
> - We added std dev in **Table 2**. The results of other baseline models are from their original paper, where the existing convention in the literature of reporting the raw scores for each game and the normalized averages.

---

> > ### Comment · Reviewer_SZnA · 2022-08-08
> > **Paper3026 Authors**
> >
> > My apologize, my previous post was not correctly posted. Thank you for the kind reminder.
> > i acknowledge the authors improvements over the paper, the small add-ons there and there maks the paper stronger, and ready for admission from my perspective. I thus increased my score. However, I also understand the other reviewer comments, and we will likely continue the discussion after the rebutal.

---

> > > ### Author Response · Authors · 2022-08-08
> > > **Thank note.**
> > >
> > > Thank you for responding to our changes and for the time you have spend in writing a thorough review with points for improvement. We are encouraged to hear that you think the paper is ready for acceptance and hope that a fruitful discussion continues into the post-rebuttal period with the other reviewers.

---

> ### Author Response · Authors · 2022-08-08
> **Gentle reminder that the discussion period closes soon**
>
> This is a gentle reminder that the discussion period closes soon. We have attempted to address all concerns raised, summarized the rationale for doing so, and modified the paper itself. We would be greatly encouraged if the reviewers either raised their scores or engaged in further discussion with us. Thanks a lot.

---

### Official Review · Reviewer_Jzdf · 2022-07-11

**Rating:** 4
**Confidence:** 3
**Soundness:** 3 good
**Presentation:** 2 fair
**Contribution:** 2 fair

**Summary:**

This paper addresses the problem of generating intermediate natural language explanations for sequential decision making in IF games. The focus is to generate immediate explanations for every step in the game and to generate temporally extended explanations to justify long-term task decisions, which depend on several intermediate steps. The approach is based on a RL Framework which uses a knowledge graph special for IF games as state representation together with several attention mechanism. To generate an immediate explanation, two steps of attention first identify the most relevant subgraph and then the most relevant KG nodes. Temporally extended explanations are generated by a cascade of filtering steps to reduce the number of relevant candidate game steps. The approach is evaluated using user studies.

**Questions:**

What would be required to generalize this approach to a related usecase, such as QA or Task-oriented Dialogue systems?

**Ethics Review Area:**

["I don’t know"]

**Limitations:**

Yes

**Strengths And Weaknesses:**

Strengths:
- The paper provides evidence that temporally-extended explanations deliver value for generating explanations.


Main Concern: Generation of temporally-extended explanations
- The generation of temporally extended explanations consists of a cascade of different components, either straightfoward statistics or prior work. It is impossible to assess the impact and rationale of each component. I encourage the authors to provide an ablation study. Using user study results, I am wondering if there is not a more general solution to assess the importance of relevant game steps to form a temporally-extended explanations. For instance, attending over all game steps to estimate P(A|B) and finetuning CALM.

---

> ### Author Response · Authors · 2022-08-02
> **Explanation model clarifications**
>
> We thank the reviewer for the time and effort and will make some clarifications below.
>
> 1. _''The generation of temporally extended explanations consists of a cascade of different components, either straightforward statistics or prior work. It is impossible to assess the impact and rationale of each component. I encourage the authors to provide an ablation study.''_
>
> - We have shown our ablation study of temporally extended explanations in Appendix C of the previous submission.
>
>     To summarize, our findings via human participant study indicate that
>     - compared to the Bayesian filter, after applying the CALM model to filter explanation candidates, generated explanations are significantly preferred by human participants;
>     - and the full HEX-RL system with Bayes+CALM+Semantic filters provided temporal explanations that human participants felt were more understandable than alternatives.
>
> 2. _''Using user study results, I am wondering if there is not a more general solution to assess the importance of relevant game steps to form temporally-extended explanations. For instance, attending overall game steps to estimate P(A|B) and finetuning CALM.''_
>
> - Thanks for your suggestion.
>
>     The reason why we did not use a **single** general model to produce temporally-extended explanations in this paper is that different filters serve different functions.
>     - Bayesian filter helps to form the most likely chain of causal dependencies that lead to a state. It only considers causal dependencies in this specific game without any semantic relationship between steps.
>     - CALM filters down the set of important states by finding the states that have corresponding actions that a human player would be more likely to perform. Since CALM is trained on human transcripts, it considers causal dependencies from the human perspective.
>     - The semantic State-Action Filter takes semantic relationships into consideration.
>
>     These three filters serve different requirements of temporally-extended explanations. One can use any combination of them for serving different functions. For example, we can produce temporally-extended explanations about causal with the Baysian filter only.
>
>     The task of combining these three functions into one model is beyond the scope of this work and we will consider it in future work.
>
> 3. _''What would be required to generalize this approach to a related use case, such as QA or Task-oriented Dialogue systems?''_
>
> - We note that the types of QA and task-oriented dialogue systems that a method like ours would be useful for would also need to be grounded in the knowledge graph of their state. I.e. there would need to be a way of extracting the knowledge graph of the world state from the input observations. In our case, this is using a QA model such as seen in Ammanabrolu et al. 2020. Once a knowledge graph of the world knowledge is present, the rest of the architecture we propose can be applied.
>
>   We elaborate on some of these requirements in the Limitation section of our previous submission. Our model is limited to providing explanations of systems affected by natural language. We added more requirements for generalizing this approach to a related use case in the *Evaluation* section of the current revision.

---

> ### Author Response · Authors · 2022-08-08
> **Gentle reminder that the discussion period closes soon**
>
> This is a gentle reminder that the discussion period closes soon. We have attempted to address all concerns raised, summarized the rationale for doing so, and modified the paper itself. We would be greatly encouraged if the reviewers either raised their scores or engaged in further discussion with us. Thanks a lot.

---

### Author Response · Authors · 2022-08-02
**Summary of changes**

We would like to thank the reviewers for their time spent reading and suggesting improvements to our work. We have made an effort to clarify every point raised by the reviewers in our revised manuscript and have detailed a summary of the changes/clarifications point by point in the text below in addition to providing a more detailed response to each reviewer. We would encourage the reviewers to consider raising their scores.

Summary of changes made by your suggestions:

- A discussion of the limitations of HEX-RL [Asked for by Reviewer Jzdf, SZnA]:
We elaborate on some limitations in the *Evaluation* section (4.2 and 4.3). Our model is limited to providing explanations of systems affected by natural language.
We added a discussion on how this method can be used for related use cases such as task-oriented dialogue or QA in the *Evaluation* section of the current revision.
In Appendix D, we also present a qualitative analysis of successful/unsuccessful explanations with example trajectories in a format similar to that shown to our human participants to give the reader a better sense of the soundness and correctness of generated explanations across baselines.

- Additional related work [Asked for by Reviewer SZnA]: We added more related works to the Related Work section and added more module names in Figure 2 to make it easier to follow in revision.

- Reproducibility details [Asked for by Reviewer SZnA]: We reported more hyperparameters of QA, A2C, and HEX-RL models in Appendix A.2, A.3, A.5, and A.9. Besides, we update performance across agents in Table 2 with standard deviations across 5 random seeds in each game (Training curves with standard deviations can be found in Appendix A.10)

- [Asked for by Reviewer H13U]: We added full action space in Appendix A.1.

---

### Author Response · Authors · 2022-08-08
**Gentle reminder that the discussion period closes soon**

This is a gentle reminder that the discussion period closes soon. We have attempted to address all concerns raised, summarized the rationale for doing so and modified the paper itself. We would be greatly encouraged if the reviewers either raised their scores or engaged in further discussion with us. Thanks a lot.

---

### Meta-Review · Area_Chair_RKj7 · 2022-08-28

**Recommendation:** Accept
**Confidence:** Certain

**Metareview:**

The paper proposes a hierarchical approach to explainable RL which combines different modules, including a knowledge graph, to generate natural language explanations.

There has been a debate between the reviewers about this approach being novel or not which was the main concern left after the rebuttal phase. Other concerns were indeed fairly well addressed by the authors.

Although each module in the proposed approach is not novel, it seems that the way they are used to address the specific problem of explainability and especially in text games is novel and sound. The results are convincing and the evaluation against a large number of baselines is the result of a large amount of work and a solid scientific method.

For these reasons, acceptance is recommended.

**Award:**

No

---

### Decision · Program_Chairs · 2022-09-14

Accept